# Acceptance of the Human Papillomavirus Vaccine among General Men and Men with a Same-Sex Orientation and Its Influencing Factors: A Systematic Review and Meta-Analysis

**DOI:** 10.3390/vaccines12010016

**Published:** 2023-12-22

**Authors:** Fang Shen, Yuxia Du, Kexin Cao, Can Chen, Mengya Yang, Rui Yan, Shigui Yang

**Affiliations:** Department of Emergency Medicine, Second Affiliated Hospital, School of Public Health, The Key Laboratory of Intelligent Preventive Medicine of Zhejiang Province, Zhejiang University School of Medicine, 866 Yuhangtang Road, Hangzhou 310058, China; 22118939@zju.edu.cn (F.S.); duyuxia@zju.edu.cn (Y.D.); 22218815@zju.edu.cn (K.C.); chencan@zju.edu.cn (C.C.); yangmengya40@163.com (M.Y.); ryan@cdc.zj.cn (R.Y.)

**Keywords:** human papillomavirus vaccine, male, men with a same-sex orientation, acceptance

## Abstract

The human papillomavirus (HPV) vaccine reduces the prevalence of genital warts and the cancers they are associated with in males. However, the vaccination of males has always been neglected. Here, we performed a meta-analysis to comprehend the acceptability of the HPV vaccine in men and the factors impacting vaccination intentions. We searched PubMed, Web of Science, Embase, Ovid, CNKI, and Wan Fang up to 5 July 2023 for studies that reported HPV vaccine acceptance among men. A random effects model was used to obtain the pooled acceptance rate, and subgroup analysis was performed. Then, the influencing factors of HPV vaccination in males were analyzed. A total of 57 studies with 32,962 samples were included in the analysis. The overall acceptance rate of the HPV vaccine in general men was 47.04% (95% confidence interval [95%CI]: 39.23–54.93%), and 62.23% (95% CI: 52.93–71.10%) among those whose sexual orientation contained men. HPV vaccine acceptance rates differed significantly between the two populations (*p* = 0.01). The population with a medical background (46.23%, 95% CI: 40.28–52.24%) was more willing to be vaccinated. In comparison to the employed population (66.93%, 95% CI: 48.79–82.81%) and the unemployed (68.44%, 95% CI: 52.82–82.23%), vaccination acceptance rates were lower among students (47.35%, 95% CI: 37.00–57.81%) (*p* = 0.04). The most significant barriers to vaccination were perceived low risk of infection for themselves (45.91%, 95% CI: 31.73–60.43%), followed by vaccine cost (43.46%, 95% CI: 31.20–56.13%). Moreover, the recommendations from medical professionals (60.90%, 95%CI: 44.23–76.37%) and sexual partners (60.09%, 95%CI: 27.11–88.67%) were significant factors in promoting vaccination. Overall, acceptance of the HPV vaccine among general men was at a lower level, despite being slightly higher among men with a same-sex orientation. Popularizing knowledge about diseases and vaccines, encouraging medical professionals to recommend vaccines to men, or reducing the cost of vaccines may promote HPV vaccination.

## 1. Introduction

Human papillomavirus (HPV) infection is one of the most common sexually transmitted infections worldwide and is transmitted through direct contact [1]. Low-risk HPV types usually cause genital warts, while high-risk HPV types are linked to several cancers in different parts of the body, including cervical, anal, vaginal, penile, and oropharyngeal cancers, which are responsible for 630,000 new cases of cancer worldwide each year [2,3]. The study found that in the United States, more than 16,500 HPV-related cancers are diagnosed in men each year, while more than 60% of penile cancer and 90% of anal cancer cases are caused by HPV [4]. In addition, HPV can also cause oropharyngeal cancer, and HPV-related oropharyngeal cancer is increasing in high-income countries, especially among men [5]. Therefore, HPV infection also poses a threat to the health of men.

The HPV vaccine can effectively prevent HPV-related diseases. The World Health Organization (WHO) recommends that both men and women should receive the HPV vaccine [6]. Studies have found that HPV vaccination can effectively prevent HPV infection as well as precancerous lesions of the anogenital tract and genital warts [7]. A meta-analysis showed a significant reduction in the incidence of HPV-related infections and diseases after HPV vaccination for 5–8 years [8]. However, the HPV vaccine is widely recognized as a female vaccine, and the vaccine was initially recommended for adolescent girls [9]. In many countries, the HPV vaccine is only included in the vaccination program for females [5]. Awareness and acceptance among males are low. Existing studies have found that the acceptance of the HPV vaccine varies among different countries and male groups [10,11]. To better understand the acceptance of the HPV vaccine and the factors impacting vaccination in the global male population, we conducted a meta-analysis. Considering that previous meta-analyses of research on men’s acceptance of the HPV vaccine covered a more restricted area of the world [12,13], we broadened our search scope and did not limit the language used in the publications. We then looked for pertinent articles on men’s acceptance of the HPV vaccine since the creation of multiple global databases, ensuring that, to the greatest extent possible, research on men’s acceptance of the HPV vaccine in a variety of nations was included in the meta-analyses. Subgroup analysis was performed on different population characteristics to explore the factors affecting the acceptance of the HPV vaccine in men.

## 2. Materials and Methods

Our study strictly followed the Preferred Reporting Items for Systematic Reviews and Meta-Analyses statement and has been registered on the PROSPERO website (CRD42023470851).

### 2.1. Definition of Study Population and HPV Vaccine Acceptance Rate

In this study, our subjects were male, regardless of age or sexual experience. Sexual orientation included men who have sex with men (MSM), gay, bisexual, and heterosexual. MSM were defined as men who have sex with men regardless of whether they self-identify as gay, heterosexual, or bisexual [14]. In each of the studies we included in our analysis, the authors used a questionnaire to investigate participants’ own willingness to be vaccinated against HPV. HPV vaccine acceptance rates are determined by dividing the number of respondents who accepted being vaccinated (or who were willing to be vaccinated) by the total number of respondents to the survey. Then we extracted the HPV vaccine acceptance rate in each included study and summarized it according to the surveyed population.

### 2.2. Search Strategy

We identified PubMed, Web of Science, Embase, Ovid, CNKI, and Wan Fang as the search databases, and the search terms included: “Human Papillomavirus Viruses”, “Human Papillomavirus”, “HPV Human Papillomavirus”, “HPV”, “Papillomavirus virus”, “Vaccines”, “immuni*”, “inoculat*”, ” hesitance*”, “willing*”, “accept*”, “perception”, “attitude”, “recognition”, “Male”, “Males”. The search strategies are listed in Appendix A. There were no time, geographic, or language restrictions on the search scope, and the last search date was 5 July 2023. Additionally, we checked the reference lists of the studies that were included and the reviews that the search turned up.

### 2.3. Inclusion and Exclusion Criteria

The original research articles published both domestically and internationally on the acceptance of or willingness to receive the HPV vaccine in men using quantitative methods were included. The articles with incomplete basic data, measurement indicators that we were unable to extract, and repeated publications were excluded.

### 2.4. Literature Screening, Data Extraction, and Quality Assessment

After deleting reviews, meta-analyses, letters, meetings, guidelines, and duplicate publications, two independent reviewers (F.S. and Y.D.) screened titles and abstracts to exclude studies that were not about men’s HPV vaccination intentions. Articles that matched the initial screening were read in their entirety to determine the final studies to be included in the analysis. When there were any disagreements during the literature screening process, the two independent reviewers talked them out or sought advice from a third reviewer (C.C.). We designed a special data extraction form using Microsoft Excel 2021, and five researchers (F.S., Y.D., K.C., M.Y., and R.Y.) independently extracted the literature information and related data. The extracted data mainly included: 1. study information (title, first author, year of publication, time of sample recruitment, study location, sample size, sexual orientation, age, marital status, educational structure, and research methods); 2. information about HPV vaccine acceptance (vaccine uptake in the study population; demographic factors: age (0–17, 18–40, 40-), sexual orientation (containing men [MSM, gay, bisexual], female only [heterosexual]), HIV status (positive or negative), medical background, educational background, partner status, history of sexually transmitted diseases, residence (rural or urban), employment status, social factors: vaccine cost (charge or free), behavioral factors: sexual experience, condom use status; 3. factors affecting HPV vaccination (barriers, promoting factors). The Agency for Healthcare Research and Quality (AHRQ) standards were used to assess the quality of the studies [15]. There were 11 entries in the cross-sectional study evaluation criteria, with the primary evaluations being the source of the data, inclusion and exclusion criteria, period of the study, selection of participants, evaluation of the subjective factors, assessment of the quality, justification for the exclusion of the data, potential confounders, treatment of the missing data, participants’ response rate, and comprehensiveness of the study. One point was assigned to each item, and studies were classified into low (0–3 points), medium (4–7 points), and high (8–11 points) according to the total score (Appendix A). Any disagreements that arose during the data extraction process or the evaluation of the study’s quality were resolved through discussion (F.S., Y.D., K.C., M.Y., R.Y., and C.C.). EndNote (version X9) was used to manage the studies and citations.

### 2.5. Statistical Analysis

A Q-test was performed to determine whether there was heterogeneity across the included articles. The degree of heterogeneity was quantitatively assessed along with the I^2^ statistic, with a fixed-effects model being selected when I^2^ was less than 25% and a random-effects model being chosen when the opposite was true [16,17]. Freeman Tukey double arcsine transformation was used to obtain the combined rate of HPV vaccine acceptance. Egger’s test and funnel plot were used to assess publication bias. Sensitivity analysis was used to evaluate the stability and reliability of the results. Subgroup analysis was used to explore the single factor influencing the combined rate. The promoting and hindering factors of HPV vaccination were analyzed according to the influencing factors of HPV vaccination provided in each study. Excel was used to collect and organize the data sets, while R 4.3.1 and RStudio were used for the meta-analysis.

## 3. Results

### 3.1. Search Results

Through a literature search, a total of 2489 documents were found. In total, 512 duplicate publications were disregarded, and 214 papers belonging to the review, meta-analysis, letter, conference, and guideline categories were also disregarded. The remaining studies were read for titles and abstracts, and 1676 articles were excluded that did not meet the inclusion criteria. The full text was read, and 45 articles were excluded due to incomplete data, non-male vaccination intention survey, and other reasons. The references of the retrieved reviews and studies that met the inclusion criteria were screened, and finally, a total of 57 articles were included in the meta-analysis (Figure 1).

The 57 articles included in the analysis were published between 2008 and 2023, and the research sites covered 17 countries on four continents. All were cross-sectional studies. A total of forty-six articles were published in English, ten in Chinese, and one in French. According to the evaluation criteria recommended by AHRQ, there were twenty-nine studies (50.88%) with high quality, twenty-seven studies (47.37%) with medium quality, and one study (1.75%) with low quality. The evaluation’s findings indicated that the majority of the scored entries were 1, 2, 3, 4, and 8, which assessed the data’s source, inclusion and exclusion criteria, research duration, participant selection, and confounding factor control. The following entries accounted for the majority of the unscored entries: (1). Item 11. The articles that were included in this study all used questionnaire surveys to obtain basic information about the study subjects, vaccination intention, and other data, none of which mentioned the follow-up of incomplete data and its subsequent results. Therefore, all the studies in this entry were classified as “unclear”. (2) Item 5. In total, 40 articles were labeled as “unclear” due to their failure to specify whether the surveys contained subjective evaluation indicators apart from the study subjects’ objective indicators. (3) Item 6. A total of 36 articles were found to be “No” for failing to describe the appropriate assessment for quality assurance. (4) Item 9. In total, 28 articles were rated as “No” because they did not provide an explanation for how missing data were handled. (5) Item 7. A total of 15 papers were judged “No” because they did not provide specific explanations for the exclusion of some subjects from the analysis. See Appendix A for details of the evaluation results of each study. The literature information is shown in Table 1.

### 3.2. HPV Vaccine Acceptance and Results of Subgroup Analysis

The total sample size of the included studies was 32,962. The pooled acceptance rate of the HPV vaccine in general men was 47.04% (95% confidence interval [95%CI]: 39.23–54.93%). In men with a same-sex orientation, the overall rate was 62.23% (95% CI: 52.93–71.10%). Figure 2 shows the sample size, weight, and respective acceptance rate of each included study. Subgroup analysis showed that there was a statistical difference in the acceptance rate of the HPV vaccine between the two groups (*p* = 0.01) (Appendix A).

According to the subgroup analysis of different characteristics and behavioral habits, the vaccination acceptance rate of people with a medical background (46.23%, 95% CI: 40.28–52.24%) was significantly higher than that of those without a medical background (33.96%, 95% CI: 26.97–41.32%) (*p* = 0.01). In comparison to the employed population (66.93%, 95% CI: 48.79–82.81%) and the unemployed (68.44%, 95% CI: 52.82–82.23%), vaccination acceptance rates were lower among students (47.35%, 95% CI: 37.00–57.81%) (*p* = 0.04). The Americas region (57.12%, 95%CI: 45.92–67.98%) and South-East Asia region (89.89%, 95%CI: 84.99–93.93%) had higher acceptance rates than the Western Pacific region (50.81%, 95%CI: 40.73–60.87%), the European region (48.09%, 95%CI: 29.63–66.82%), and the Eastern Mediterranean region (47.54%, 95%CI: 44.23–50.86%). Appendix A shows the sample size of studies and combined acceptance rates for countries in the different WHO regions. Furthermore, between different ages (0–17 [53.40%] vs. 18–40 [60.03%] vs. 40~ [61.67%]), areas of residence (rural 58.03% vs. urban 65.17%), educational backgrounds (high school and below 58.87% vs. college and above 65.70%), and partner statuses (yes 60.42% vs. no 64.40%), there was no statistical difference (*p* > 0.05) (Figure 3).

As illustrated in Figure 4, for sexual experience (yes 44.77% vs. no 38.97%), history of sexually transmitted diseases (yes 74.83% vs. no 66.77%), HIV status (positive 73.47% vs. negative 78.40%), condom use (yes 61.52% vs. no 63.96%), and cost of the vaccine (free 73.51% vs. charge 47.15%), though the acceptance rates were different, they did not show statistical differences (*p* > 0.05).

Acceptance of the HPV vaccine among men showed variations over time, with the highest acceptance in the 2004–2005 period (66.55%) and the lowest in 2010–2011 (32.29%). However, the difference was not statistically significant (*p* = 0.11). Changes in men’s acceptance of the HPV vaccine over time are displayed in Appendix A.

### 3.3. Analysis of Factors Influencing HPV Vaccination Intention

Recommendations from healthcare providers (60.90%, 95%CI: 44.23–76.37%) and sexual partners (60.09%, 95%CI: 27.11–88.67%) and the perceived risk of disease (54.52%, 95%CI: 28.16–79.64%) were important factors in promoting vaccination. Free vaccination or coverage by medical insurance (53.28%), protective partners (53.15%), and recommendations from friends (30.90%) were also the main reasons for increasing the willingness to vaccinate (Figure 5A).

The most significant barriers to vaccination were perceived low infection risk (45.91%, 95% CI: 31.73–60.43%) and vaccine cost (43.46%, 95% CI: 31.20–56.13%). Concerns about the vaccine’s safety (31.37%, 95% CI: 16.99–47.82%) and effectiveness (30.06%, 95% CI: 18.42–43.15%) and fear of unfavorable side effects (30.52%, 95% CI: 17.02–45.97%) were also significant deterrents among men. In addition, the belief that they do not need the vaccine (28.23%), embarrassment because the HPV vaccine is a preventive against sexually transmitted diseases (22.13%), belief that the HPV vaccine causes sexual promiscuity (21.17%), the need to complete three doses within six months (19.94%), distrust of the vaccine (17.84%), fear of needles (13.17%), transportation (16.24%), and time expenses (13.10%) were also reasons for refusal of vaccination (Figure 5B).

### 3.4. Publication Bias and Sensitivity Analysis

The shape of the funnel plot was relatively symmetrical (Appendix A). Egger’s test was performed, and the *p* value was 0.94 (*p* > 0.05), indicating that no significant publication bias was found. The leave-one-out method was used to conduct a sensitivity analysis of the included tudies. By excluding each of the studies one by one, the rate of the remaining studies was pooled. The results showed that the combined rates obtained by excluding each of the studies one by one did not show a large difference: all of them were in the range of 45.72–48.25% and did not present a large change compared with the total combined rate of 47.03% (Appendix A). This indicates that the pooled results of the overall HPV vaccine acceptance rate are robust and reliable.

## 4. Discussion

The results showed that the overall acceptance of the HPV vaccine in general males was 47.04%, lower than in men with a same-sex orientation (62.23%). This indicated that males of different sexual orientations had varying levels of acceptance for the HPV vaccine. Those whose sexual orientation included men were more likely to be accepted than those whose orientation just included women. This contradicts the conclusions made by Newman et al. [12] in 2013. This may be due to the larger sample size and wider population included in this study. Men who are gay, bisexual, or MSM are more likely to engage in risky sexual conduct and are more aware of HPV-related sexually transmitted diseases (STDs) and vaccine information than the general public [27]. In addition, studies have shown that people whose sexual orientation includes men are less protected as a group than those whose sexual orientation includes only women and that the benefits of vaccination are greater for these groups [73,74]. Heterosexual men can gain from the female population’s immunization as a whole, and their individual vaccination can also protect their partners, particularly if administered before they start having relations [74].

The acceptance rates of general men and men with a same-sex orientation were both lower than that found by Dursun et al. in a review of HPV vaccine acceptance in the female population (70%) [75]. The lower acceptance of the HPV vaccine in men than in women may be related to the early promotion of the HPV vaccine to women, the widely recognized fact that persistent HPV infection can cause cervical cancer in women [13], and the lack of knowledge that men are susceptible to genital warts and related cancers after HPV infection.

The findings demonstrated that the acceptance rate was greater among employed and unemployed individuals than it was among students. This implies that, compared to the social population, students had a lower acceptance rate for the HPV vaccine. Studies have found that students typically believe they are at a lower risk of acquiring HPV and other STDs than society [3,6]. This is a sign that one potential strategy to raise the perceived risk of HPV and vaccine uptake among students is through school-based health education. It is worth noting, however, that the younger unemployed population may overlap with students in terms of age and that this age-related confounding factor may have an impact on the population’s actual acceptability. Future research should focus on the impact of age on various professional groups’ acceptance of the HPV vaccine. Men with a medical background had a higher willingness to receive the HPV vaccination, suggesting that knowledge of HPV and HPV vaccines could promote vaccination [6,25,65]. Previous studies have also found that after information popularization about HPV and its vaccine was carried out for people from other professional backgrounds, there was no difference in HPV vaccine acceptance compared with people from medical backgrounds [68]. In addition, there were differences in the acceptance of the vaccine among countries in different WHO regions. The acceptance rates in the Americas region and the South-East Asia region were higher than those in the Western Pacific region, European region, and Eastern Mediterranean region. The meta-analysis showed a high acceptance rate of 89.89% in South-East Asia. This may be related to the small sample size and number of studies obtained from the region (only one study in one country), which makes the results biased. In the Americas region, the literature included in this study had the highest number of studies on HPV vaccine acceptance among men in the United States. The U.S. FDA approved the quadrivalent HPV vaccine in 2006 [18], making it the first country in the world to promote HPV vaccination. The Western Pacific region is preceded only by the Americas, then Europe and the Eastern Mediterranean region. Studies of HPV vaccine uptake in men vary by region and country. Meanwhile, there is a general lack of representation due to the scarcity of pertinent research in various fields. This shows that additional research should be carried out to support the study on the acceptance rate of the HPV vaccine in male populations worldwide, particularly in the Eastern Mediterranean, South-East Asian, and African regions.

This study discovered that a key element in boosting immunization was the recommendation of a healthcare provider. Most men (60.90%) said they were more likely to get the HPV vaccine at their doctors’ advice. Men are encouraged by physicians’ suggestions [71]. However, the fact that healthcare providers are more likely to recommend the HPV vaccine to women than men demonstrates that healthcare providers need to raise awareness of HPV vaccination in men as well [71,76]. Furthermore, recommendation from sexual partners, perceived risk of disease, free vaccination or coverage of the vaccine by medical insurance, and recommendation from friends were also the main factors contributing to men’s willingness to receive the vaccine. In conclusion, support from partners and friends, high awareness of HPV and its vaccine, and the coverage of the HPV vaccine in the national immunization program or medical insurance can improve the acceptance of HPV vaccine in the male population [6,19].

Among the men who refused to be vaccinated, nearly half (45.91%) believed that they were not at risk of HPV infection, which reflected the lack of knowledge about HPV infection among men. The misconception that HPV is “feminine” leads to gender bias at the level of public awareness [61,77]. It is urgent to strengthen the awareness of human papillomavirus in the male population, popularize the knowledge of the disease, and improve public awareness of self-protection. In addition, the high cost of the vaccine is also the main reason for males to refuse to be vaccinated. Although the HPV vaccine has been introduced into the immunization programs of more than 100 countries around the world, many countries do not include men due to the limitations of the applicable population [5]. This means that men have to pay for the vaccine out of their own pockets, and the high cost leads people who are less able to afford it to choose to refuse the vaccine. In this study, the subgroup analysis of vaccine cost (free vs. charge) found that there was a large difference in the acceptance rate of free and self-funded vaccines in the male population (73.51% vs. 47.15%, *p* = 0.07). The statistically insignificant difference may be related to the small number of relevant studies included. Further investigation and inquiry are needed in the future. Furthermore, the safety, efficacy, and side effects of the vaccine, thinking that they do not need the vaccine, feeling embarrassed because the HPV vaccine is a prevention of sexually transmitted diseases, thinking that the HPV vaccine will cause sexual promiscuity, need to complete three doses within 6 months, distrust of the vaccine, fear of injections, and the cost of transportation and time were also reasons for male respondents to refuse the vaccination. It is necessary to carry out disease and vaccine knowledge popularization campaigns, provide HPV-related education through authoritative sources [78], and publish the vaccine authorization process and post-marketing surveillance data to the public [69] to improve the awareness of HPV and the trust in the vaccine.

There are several limitations to our study. First, the included literature did not study a broad enough area. Although we tried our best to expand the scope of inclusion and the search of the database was not limited by time, geography, or language, there is still a lack of relevant data on the acceptance of the HPV vaccine among men in many countries, especially in Oceania and Africa, which means that the universality of HPV vaccine acceptance among men in the world may be limited. Second, nearly half of the included studies were of medium quality, and another one was of low quality. Future studies should be designed more strictly to improve the quality of research. Third, high heterogeneity was found when the results were pooled. The included studies were from all over the world, and due to the large differences in economic levels and culture in different countries, different male groups may have different vaccination intentions. Moreover, all the included studies were cross-sectional studies, and the data were obtained through on-site questionnaires, which may cause information bias. All of these factors may lead to heterogeneity in the pooled results.

## 5. Conclusions

In conclusion, acceptance of the HPV vaccine among general men was at a lower level, despite being slightly higher among men with a same-sex orientation. In general, vaccine refusal and hesitancy were caused by a lack of knowledge about the disease, a lack of knowledge about the vaccine, and a high cost. In the future, more extensive publicity and health education should be carried out to the public through all kinds of authoritative media and units to popularize the knowledge of HPV and its vaccine. In addition, measures such as encouraging medical professionals to recommend vaccines to men or reducing the cost of HPV vaccination could increase the willingness of males to receive HPV vaccination.

## Figures and Tables

**Figure 1 vaccines-12-00016-f001:**
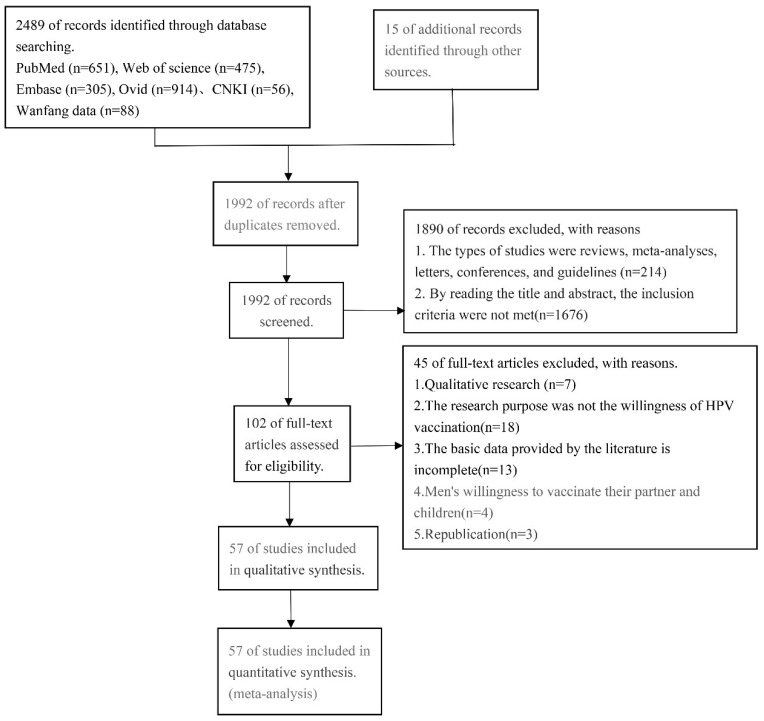
Flow diagram of article selection for human papillomavirus vaccine acceptability among men.

**Figure 2 vaccines-12-00016-f002:**
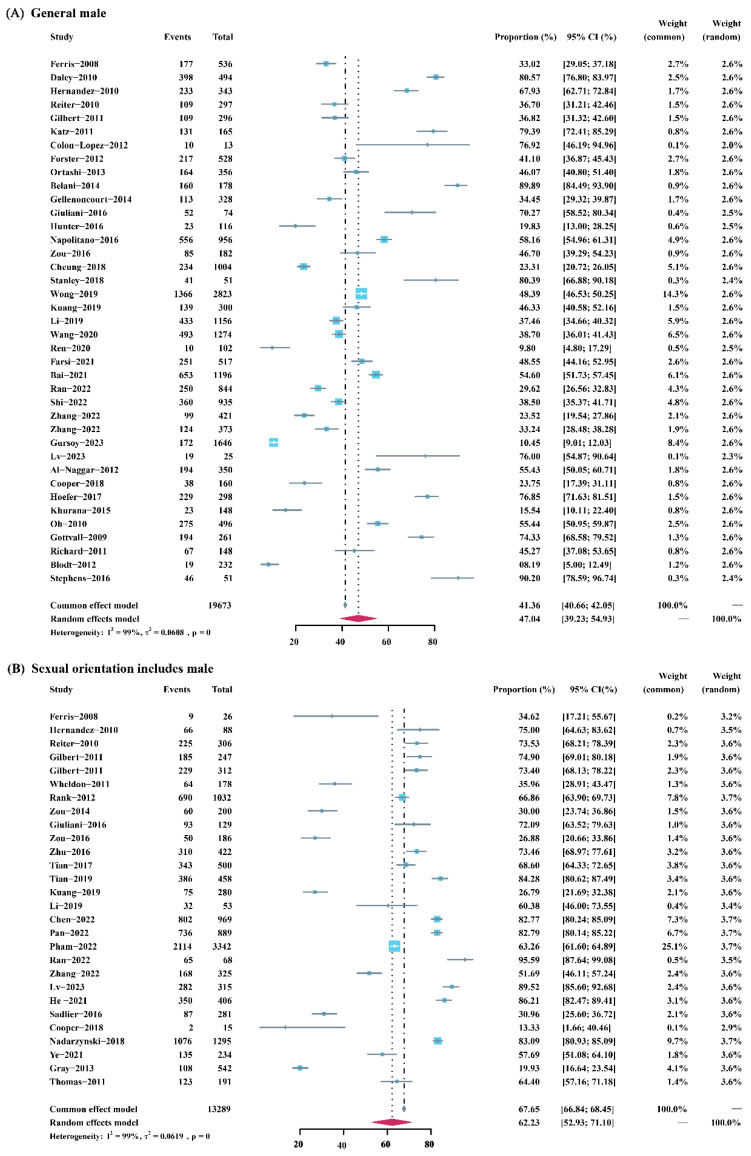
Forest plot of meta-analysis for HPV vaccine acceptance among general men and men with a same-sex orientation.

**Figure 3 vaccines-12-00016-f003:**
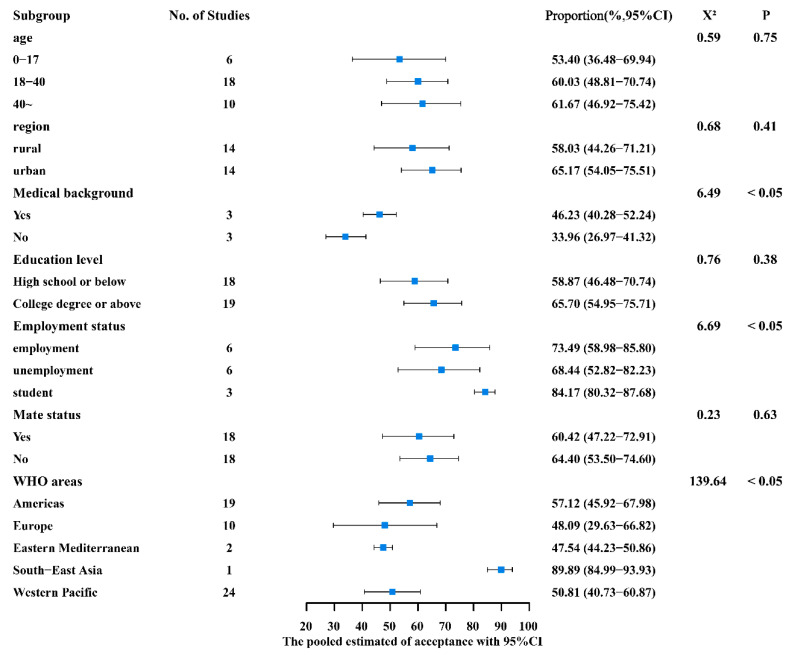
Subgroup analysis of basic demographic characteristics.

**Figure 4 vaccines-12-00016-f004:**
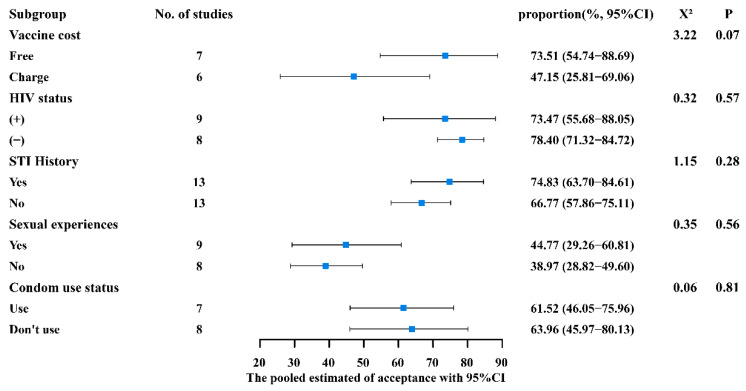
Subgroup analysis of general population conditions.

**Figure 5 vaccines-12-00016-f005:**
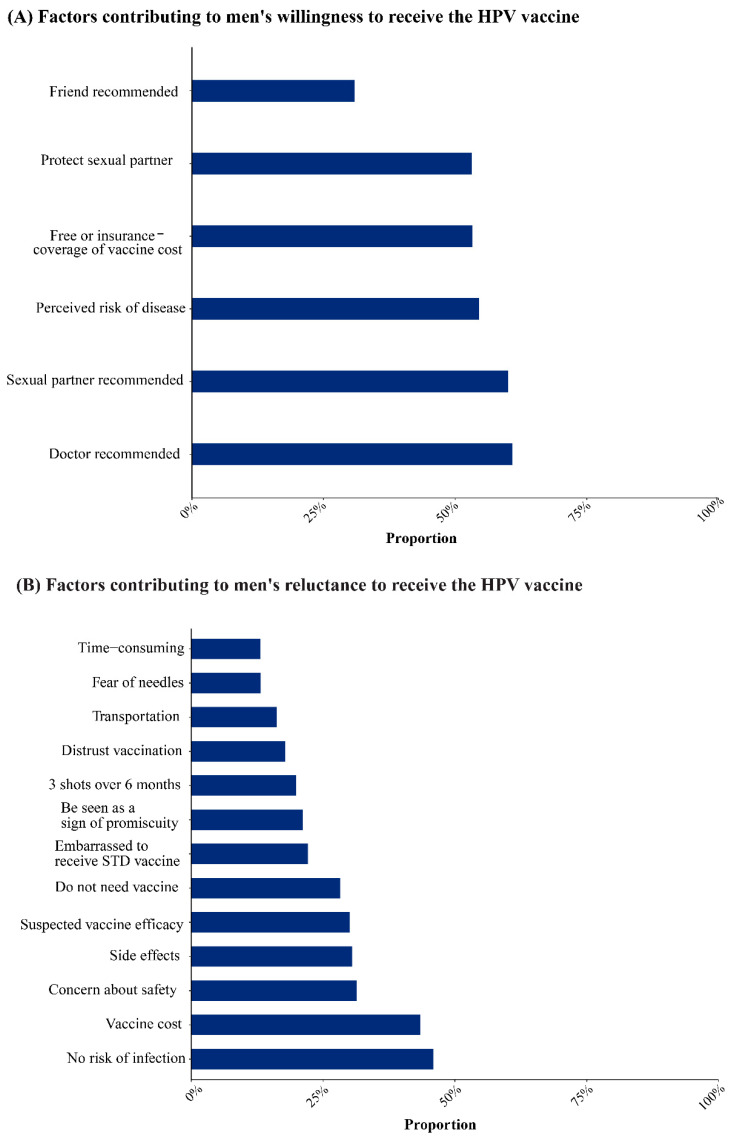
Factors influencing HPV vaccination in men.

**Table 1 vaccines-12-00016-t001:** Basic information of all articles included.

Author	Year	Age	Survey Time	Country	Sample Size	Sexual Orientation/Behavior	Research Methods	Acceptance Rate (%)	Quality Assessment *
Ferris, D., et al. ^†^ [18,19]	2008	18–45	—	the U. S	571	heterosexual/ homosexual/bisexual	cross-sectional study	32.92	High
Gottvall, M., et al. [20]	2009	15–16	2008 Autumn	Sweden	261	—	cross-sectional study	74.33	High
Daley, E., et al. [21]	2010	18–70	March 2007	the U. S	494	—	cross-sectional study	80.57	High
Hernandez, B. Y., et al. [22]	2010	18–79	2004–2007	the U. S	445	MSM/heterosexual	cross-sectional study	69.21	Moderate
Reiter, P., et al. [23]	2010	18–59	January 2009	the U. S	306	homosexual/bisexual	cross-sectional study	73.53	High
Reiter, P. L., et al. [24]	2010	18–59	January 2009	the U. S	297	heterosexual	cross-sectional study	36.70	High
Oh, J. K., et al. [25]	2010	≥20	November 2007	Korea	496	—	cross-sectional study	55.44	High
Gilbert, P., et al. [26]	2011	—	January 2009	the U. S	247	homosexual	cross-sectional study	74.90	High
Gilbert, P., et al. [27]	2011	18–59	January 2009	the U. S	608	heterosexual/bisexual/homosexual	cross-sectional study	55.59	High
Katz, M. L., et al. [28]	2011	16–26	2009	the U. S	165	—	cross-sectional study	79.39	Moderate
Wheldon, C. W., et al. [29]	2011	18–29	September 2010–December 2010	the U. S	179	homosexual/bisexual	cross-sectional study	35.75	High
Thomas, Emily A., et al. [30]	2011	36.57 ^‡^	June 2009–September 2009	the U. S	191	MSM	cross-sectional study	64.40	Moderate
Richard A., et al. [31]	2011	18–24	—	the U. S	148	—	cross-sectional study	45.27	Moderate
Colon-Lopez, V., et al. [32]	2012	18–26	2009–2010	the U. S	46	MSM/heterosexual	cross-sectional study	76.92 ^§^	Moderate
Forster, A. S., et al. [33]	2012	16–18	March2009 and September 2009	the U. K	528	—	cross-sectional study	41.10	High
Rank, C., et al. [34]	2012	19–83	July 2008–February 2009	Canada	1041	homosexual/bisexual/ other	cross-sectional study	66.95	High
Bourke, L. [35]	2012	18–27	November 2011–March 2012	Malaysia	350	heterosexual/ homosexual/none	cross-sectional study	55.43	Moderate
Blodt, S., et al. [36]	2012	18–25	July 2010	Germany	245	—	cross-sectional study	8.19 ^§^	Moderate
Ortashi, O., et al. [37]	2013	21 ^‡^	June 2012–August2012	U.A. E	356	—	cross-sectional study	46.07	Moderate
Gray, Clive M., et al. [38]	2013	18–60	September 2010–January 2011	China	542	homosexual/bisexual/ heterosexual	cross-sectional study	19.93	High
Belani, H. K., et al. [39]	2014	18–45	November 2010 and February 2011	India	178	—	cross-sectional study	89.89	Moderate
Gellenoncourt, A. and P. Di Patrizio [40]	2014	16–18	May 2013–June 2013	France	328	—	cross-sectional study	34.45	High
Zou, H. C., et al. [41]	2014	16–20	September 2010–August 2012	Australia	200	MSM	cross-sectional study	30.00	Moderate
Khurana, S., et al. [42]	2015	11–21	April 2011–February 2012	the U. S	154	—	cross-sectional study	15.54 ^§^	High
Giuliani, M., et al. [43]	2016	33 ^‡^	April 2013–June 2013	Italy	423	MSM/MNSM	cross-sectional study	34.28	High
Hunter, T. and M. Weinstein [44]	2016	18–26	—	the U. S	116	—	cross-sectional study	19.83	Moderate
Napolitano, F., et al. [45]	2016	14–24	January 2015–April 2015	Italy	956	—	cross-sectional study	58.16	High
Zou, H. C., et al. [46]	2016	≥18	April 2014–July 2014	China	368	MSM/MNSM	cross-sectional study	36.68	Moderate
Zhu, J, H., et al. [47]	2016	16–72	March 2014–September 2014	China	422	MSM	cross-sectional study	73.46	Moderate
Sadlier, C., et al. [48]	2016	≥18	January 2014–April 2014	Ireland	302	MSM	cross-sectional study	30.96 ^§^	Moderate
Stephens, Dionne P., et al. [49]	2016	18–24	—	the U. S	51	—	cross-sectional study	90.20	Moderate
Tian, T. [50]	2017	18–65	April 2016–October 2016	China	500	homosexual/bisexual	cross-sectional study	82.00	High
Hoefer, Lea, et al. [51]	2017	18–55	July 2015.7–October 2015	Greece	298	—	cross-sectional study	76.85	Moderate
Cheung, T., et al. [52]	2018	16–29	June 2015–September 2015	China	1004	—	cross-sectional study	23.30	High
Stanley, C., et al. [53]	2018	—	January 2016–May 2016	Canada	63	—	cross-sectional study	80.39 ^§^	High
Cooper, Dexter L., et al. [54]	2018	18–27	—	the U. S	190	heterosexual/ sexual minority	cross-sectional study	21.05	Moderate
Nadarzynski, T., et al. [55]	2018	14–63	July 2015–September 2015	the U. K	1508	MSM	cross-sectional study	83.02	Moderate
Tian, T., et al. [56]	2019	—	—	China	458	MSM	cross-sectional study	84.28	Moderate
Wong, L. P., et al. [57]	2019	13–14	February 2013–April 2013	Malaysia	2823	—	cross-sectional study	48.39	High
Kuang, J, Y., et al. [58]	2019	≥18	March 2018–March 2019	China	580	MSM/MNSM	cross-sectional study	36.90	low
Li, Y, F., et al. [59]	2019	21 ^‡^	March 2018–June 2018	China	1209	homosexual/bisexual/ heterosexual	cross-sectional study	38.46	High
Wang, S., et al. [3]	2020	16–26	September 2018–December 2018	China	1274	—	cross-sectional study	38.70	High
Ren, W, J., et al. [60]	2020	22–35	—	China	102	—	cross-sectional study	9.80	Moderate
Farsi, N. J., et al. [61]	2021	21 ^‡^	2017–2018	Saudi Arabia	517	—	cross-sectional study	48.55	High
Bai, M, M [62]	2021	18–26	August 2019–December 2019	China	1196	heterosexual/bisexual/homosexual	cross-sectional study	54.60	High
Wei He, et al. [63]	2021	≥18	—	China	406	MSM	cross-sectional study	86.21	High
Ye, Ze-Hao, et al. [64]	2021	≥18	July 2020–December 2020	China	234	MSM	cross-sectional study	57.69	Moderate
Chen, Q., et al. [65]	2022	14–55	—	China	969	MSM	cross-sectional study	82.77	Moderate
Pan, H., et al. [66]	2022	16–45	June 2021	China	889	MSM	cross-sectional study	82.79	High
Pham, D., et al. [67]	2022	44 ^‡^	—	the U. S	3342	homosexual/bisexual/ heterosexual/other	cross-sectional study	63.26	Moderate
Ran, H., et al. [68]	2022	≥18	February 2021–May 2021	China	912	heterosexual/ homosexual/bisexual	cross-sectional study	34.54	High
Shi, J., et al. [6]	2022	16–26	September 2019–November 2019	China	935	—	cross-sectional study	38.50	Moderate
Zhang, H, et al. [69]	2022	19.38 ^‡^	April 2021	China	421	—	cross-sectional study	23.52	High
Zhang, J., et al. [70]	2022	18–72	August 2016–June 2019	China	711	heterosexual/ homosexual/bisexual	cross-sectional study	42.33	Moderate
Gursoy, M. Y. and F. Sagtas [71]	2023	18–38	February 2022–April 2022	Turkey	1723	heterosexual/ homosexual/bisexual	cross-sectional study	10.45 ^§^	Moderate
Lv, H, W., et al. [72]	2023	≥18	July 2022–August 2022	China	407	heterosexual/ homosexual/bisexual	cross-sectional study	88.53 ^§^	Moderate

* Study quality assessment is specified in Appendix A; ^†^ Two articles by the same author and sample; ^‡^ Median; ^§^ Vaccination intention surveys do not include the full sample size; MSM, men having sex with men; MNSM, men not having sex with men.

## Data Availability

All data were included in this manuscript.

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
