# Peer review of "Acceptance of the Human Papillomavirus Vaccine among General Men and Men with a Same-Sex Orientation and Its Influencing Factors: A Systematic Review and Meta-Analysis"

_vaccines, 2023, doi:10.3390/vaccines12010016_

Round 1

Reviewer 1 Report

Comments and Suggestions for Authors

Shen et al. conducted a literature meta-analysis to assess the acceptability of the HPV vaccine in men and explore the factors influencing vaccination intentions. The manuscript is noteworthy for its clear and professional use of the English language and well-organized tables. The authors have also developed a comprehensive discussion section that thoroughly examines the factors affecting vaccination intentions. Furthermore, the manuscript reviews up-to-date literature and includes a current list of references. I have a few comments below that may enhance the clarity of this manuscript:

(1) The abbreviation "HPV" is commonly used and appears frequently in the literature. Could you please specify if the abbreviation "HPV" itself is one of the search terms?

(2) It might be worthwhile to cite table S2 when describing "quality assessment," as listed in table 1. Additionally, in table S2, regarding literature quality assessment, could you provide details on what items are represented by items 1 through 11?

(3) When concluding that "employed individuals and the unemployed were more likely to accept the vaccine than students," did you consider the factor of age? Also, are there overlaps between unemployed individuals and students? Please provide clarification.

(4) The use of "men whose sexual orientation contained men" in the title and elsewhere is a bit awkward to read. The authors may want to consider alternative ways of expressing this idea, such as "men with a same-sex orientation."

Reviewer 2 Report

Comments and Suggestions for Authors

Estimated Authors, 

I've read with great interest the present systematic review and meta-analysis from Shen et al. Authors, by collecting and summarizing a total of 57 papers from a Global perspectives reported and discussed on the acceptance of HPV vaccination and main root causes of acceptance/refusal in men/MSM.

The study is therefore interesting from a public health point of view, well documented, and well designed.

In fact, I will recommend its acceptance after minor amendments that could be summarized as follows:

1) Authors have performed quality check of collected studies and provided a sensitivity analysis. Both topic are referenced but not accurately discussed across the main text, with the reader forced to check across the supplementary material. While the choice for including the material only as supplementary can be shared, please provide a more extensive discussion of both quality check and sensitivity analysis. In this regard, Authors should explain how sensitivity analysis was performed: I guess "leave-one-study-out" approach, but it should be explained.

2) As you've accurately discussed, the geographic representativity of the meta-analysis is limited; an interesting iteration could be providing at least as supplementary material a summary table with the following informative information (only in descriptive terms): a) number of studies from WHO areas, sampled population, acceptance rate; b) studies by countries and sampled population, acceptance rate.

3) please provide an estimate of the acceptance rate across the time, in order to ascertain whether it remained stable or improved over time.

Comments on the Quality of English Language

The paper is well written. I've no specific requests.

Reviewer 3 Report

Comments and Suggestions for Authors

The topic has been extensively studied in the literature, and many papers have investigated it. However, the statement "In view of the previous meta-analysis on the acceptance of the HPV vaccine in men [12, 13], we expanded the search scope, removed all geographical, linguistic, and time restrictions, and included as many papers on the acceptance of the HPV vaccine in men from as many different countries as we could" doesn't make sense. This is because eliminating the determinants of vaccine attitude reduces our understanding of the phenomena.

Reviewer 4 Report

Comments and Suggestions for Authors

I do not have any significant comments toi give about the introduction, the method used, the result presentation.

I would like just the authors to review their analysis about the geographic factors. (the use of generic identity for Asia, Europe..)

The population groups in these regions are very heterogenous while the surveys are usually focused . Asian countries are very limited ( India, Indonesia Vietnam...) as well as European ( Northern and Eastern European countries , for exemple, are not represented)

Better not to "generalize"the differences between the regions, while the recommendations could apply for most public health policy makers in countries over the world.

Round 2

Reviewer 3 Report

Comments and Suggestions for Authors

-----